# Coordinated Navigation of Two Agricultural Robots in a Vineyard: A Simulation Study

**DOI:** 10.3390/s22239095

**Published:** 2022-11-23

**Authors:** Chris Lytridis, Christos Bazinas, Theodore Pachidis, Vassilios Chatzis, Vassilis G. Kaburlasos

**Affiliations:** 1HUMAIN-Lab, International Hellenic University (IHU), 65404 Kavala, Greece; 2Department of Management Science & Technology, International Hellenic University (IHU), 65404 Kavala, Greece

**Keywords:** Gazebo simulation, agricultural robot, cooperative navigation, autonomous navigation

## Abstract

The development of an effective agricultural robot presents various challenges in actuation, localization, navigation, sensing, etc., depending on the prescribed task. Moreover, when multiple robots are engaged in an agricultural task, this requires appropriate coordination strategies to be developed to ensure safe, effective, and efficient operation. This paper presents a simulation study that demonstrates a robust coordination strategy for the navigation of two heterogeneous robots, where one robot is the expert and the second robot is the helper in a vineyard. The robots are equipped with localization and navigation capabilities so that they can navigate the environment and appropriately position themselves in the work area. A modular collaborative algorithm is proposed for the coordinated navigation of the two robots in the field via a communications module. Furthermore, the robots are also able to position themselves accurately relative to each other using a vision module in order to effectively perform their cooperative tasks. For the experiments, a realistic simulation environment is considered, and the various control mechanisms are described. Experiments were carried out to investigate the robustness of the various algorithms and provide preliminary results before real-life implementation.

## 1. Introduction

The field of agricultural robotics is an emerging research field that has gained attention in recent years because of its potential for improving and automating agricultural work and also addressing some of the problems that exist in traditional agricultural practices, such as labor shortages. The aim of this research field is the development of fully autonomous machines that perform agricultural tasks reliably. Extensive work has been carried out that has produced both experimental prototypes as well as commercial products [1,2,3,4]. For an agricultural robot to be effective in carrying out a task autonomously, several operational aspects have to be addressed, including localization, navigation, sensing, actuation, communications, etc.

In particular, navigation is an area that requires careful consideration due to the particular characteristics of outdoor agricultural environments. Navigation requires environmental perception and control algorithms. In terms of perception, mobile robots need to sense their position in the environment and move towards areas of interest while avoiding obstacles [5,6]. This has been achieved using a variety of sensors, including satellite positioning systems, laser sensors, inertial measurement units (IMU), vision sensors, etc. Light-detecting and ranging (LiDAR)-sensors are very commonly used for the navigation of robotic systems because of their accuracy and the range of measurements. These sensors use laser beams to scan the surrounding environment, detect environmental features, and measure distances. The analysis of these features helps determine the position of the robot and identify regions of interest. The advantage of LiDAR sensors is their accuracy and fast sampling rate. In particular, agricultural robots have been used to determine positioning with respect to the target crops. Moreover, it is a common scenario in agricultural applications that the robot is required to navigate within crop rows, and for this to be achieved, it requires special considerations [7]. A recent work [8] describes the use of a 2D LiDAR to detect a local feature using DBSCAN to update the robot’s position using a particle filter and then build a path with a Kalman filter. In [9], a simulation study with an agricultural robot and LiDAR-based navigation is presented. The robot uses a point cloud retrieved from the 2D LiDAR for phenotyping tasks and also to maintain the robot’s parallel equidistance to the crop rows. In [10], a particle filter and a Kalman filter were used to determine angular and lateral deviations from the center line in an apple orchard row based on measurements from a LiDAR module.

Other localization methods have also been proposed, utilizing alternative sensors such as cameras, inertial measurement units (IMU), and GPS, often in conjunction with sensor-fusion algorithms, in order to improve accuracy. For example, a rice transplanter able to generate safe routes using an IMU unit in a previously GPS-mapped paddy field was introduced in [11]. A robot utilizing an RTK GPS navigation module for precise navigation in potato plantations is presented in [12]. A Kalman filter in conjunction with IMU and vision data was utilized for pose estimation in [13] when given a recognized object captured by a camera. Xue, Zhang, and Grift have used a vision-based method to detect the location of corn plants in a variable field of view and thus determine the location and heading of the robot [14]. 

Cooperative navigation for autonomous agricultural vehicles has also been considered in agricultural robots [15,16,17,18]. Cooperative approaches in ground vehicles come in two main forms: (a) where the workload is spread across several autonomous homogeneous machines that perform the task in a coordinated parallel fashion and, therefore, achieve the completion of a task faster, and where (b) different roles are assigned to multiple heterogeneous machines that have different capabilities which work cooperatively towards the effective completion of the task. Many studies have implemented a leader-follower architecture. For example, one of the earliest works can be found in [19]. A motion control algorithm is applied where the leader robot directs the follower robot to visit specific locations (while avoiding collisions with the leader robot) or to mimic the leader robot’s motion. In [20], two robot tractors coordinate so that they choose their followed path in order to avoid collisions and improve their field coverage. In this paper, each robot retrieves pose information from its sensors and a stored map and then coordinates its motion in the field using communication based on a server/client architecture. The motion was coordinated by ensuring a safe distance between the robots. The results of the simulations showed that it is possible for two robots to operate safely and with higher efficiency than when using a single robot. More recent work, presented in [21], showed a dual master/slave architecture employed to drive a harvesting robot and a transport robot. The system uses GNSS to determine which navigation method (GNSS-based or LiDAR-based) the robots use. The waypoints are then determined. The authors validate their approach through orchard trials, where it was demonstrated that the observed communication losses and position errors are acceptable. In [22], a framework for task assignment and trajectory planning was studied for multiple agricultural machines. Tasks were assigned to each robot either randomly or based on an ant colony-inspired calculation, which involves taking into account supply and demand, the operational capacity of the robots, and path costs. The work in [23] proposes the use of fuzzy sets to implement ontological information exchange toward multirobot coalition formation during the simulation of a precision agriculture scenario. It was shown that information exchange could increase the variability of formed coalitions and can result in a better understanding of robot behavior by humans. The effect of using controlled versus uncontrolled traffic systems for the coordination of two heterogeneous robots (an application unit and a refilling unit) on field efficiency was investigated in [24]. The results show that using controlled traffic farming increases the distance traveled by the application unit. Two combines that communicate their position and work status to each other in order to coordinate rice harvesting and avoid collisions were developed in [25]. In this case, the two combines harvest the field concurrently, yet independently, by following independent paths in counter-clockwise spiral shapes at some distance apart. On the other hand, in [26], a graphical theoretic approach was considered to plan and optimize the paths for robots in a heterogeneous robot team, where larger areas of the field were assigned to the more capable robots in the team for coverage during a monitoring task. Other approaches focus on the determination of the most efficient field coverage by multiple robots [27,28,29,30,31]. These approaches consider minimalistic robots that are either controlled by a central controller or their control is completely distributed. The objective of this type of work is to minimize the cost of field traversal or maximize the information gathered by the robot team. 

It can be seen from the above that most work in the literature deals with field coverage applications in agricultural scenarios where unmanned ground vehicles are considered. In most cases, the use of multiple robots serves as a means to increase the efficiency of the task, which, otherwise, would be carried out by a single robot. However, few papers focus on tasks where the robots need to actively cooperate (via physical interaction) in order to complete the task. In these cases, the navigational strategy needs to be adjusted so that it allows for the execution of the task itself.

The work described in this paper is part of an R&D project acknowledged in the “Funding“ note below, in which robots are engaged in various viniculture tasks. Agricultural robots, or agrobots for short, are considered to be cyber-physical systems (CPS) with sensing capabilities [32], in which software components control the mechanisms involved in carrying out skillful agricultural tasks, such as pruning, harvesting, spraying, and others, with each carried out by single or multiple cooperating robots. In the grape harvesting scenario examined in this paper, two cooperating robots are required. The focus is on a robust collaborative navigation strategy in a collaborative harvesting scenario in a vineyard. In this scenario, two robots: an expert robot responsible for performing harvesting, and a helper robot, tasked with supporting the expert robot by carrying and transporting the harvested grapes, navigate collaboratively in a vineyard. When inside a vineyard row, the expert robot begins harvesting and deposits the harvested grapes to a basket located at the back of its chassis. When the basket is full, the helper robot picks up the basket and deposits its contents into a larger basket which, if full, needs to be emptied at another location while the expert robot remains at its current position. For this scenario to be possible, the following criteria must be met: (a) the robots must accurately navigate the vineyard, and (b) the robots must be able to position themselves in such a way that precise manipulation is possible. In this paper, well-established localization and motion planning methods, communication capabilities, and task-specific navigation modules were employed to form a novel modular collaborative strategy that allows both criteria to be satisfied. The validity of the proposed collaborative strategy is tested via a realistic simulation.

More specifically, the robots possess the sensing and navigation capabilities to navigate autonomously within predetermined paths, but they are also able to appropriately position themselves in vineyard rows using LiDAR sensors and a predefined map. The robots use communication to coordinate their movements inside the vineyard. The relative positioning of the robots for the actual harvesting is achieved in two steps: first, the helper robot approaches the expert using its positioning sensors. Second, the relative positioning is corrected using vision. This ensures that the helper robot will be able to successfully manipulate the basket stored on the expert robot. The aforementioned actions are achieved through the execution of a set of selfcontained navigation algorithms. Therefore, in summary, the main contribution of this paper is the modular coordination strategy which achieves the effective coordinated navigation of the two robots (in a vineyard situation) toward the completion of a cooperative harvesting task. More specifically, the proposed strategy involves the appropriate localization and planning algorithms as well as positioning methods that enable the execution of the cooperative harvesting task, which involves manipulation between the two robotic platforms. The strategy dictates how the algorithms executed in each robot achieve coordinated navigation and enable the manipulation function. The simulation environment serves as the testbed for the proposed strategy.

The paper is structured as follows: Section 2 presents the simulation environment and the localization and navigation methods employed for the two robots. In Section 3, the results of the simulation experiments that were carried out in order to demonstrate the validity of the proposed collaborative navigation strategy are presented. Finally, in Section 4, the simulation results are discussed, and in Section 5, the conclusions of this work are outlined, and proposals for future implementations are made.

## 2. Materials and Methods

### 2.1. Simulation Environment

The simulation was developed within the open-source Gazebo simulation environment. Gazebo is the standard for the simulation of virtual environments and has a strong integration with the ROS environment. It features a physics engine for studying the dynamics of the robotic system and provides accurate visualization. To recreate a realistic virtual vineyard, a 3D model of a vineyard was utilized. Vineyard rows were created by positioning repeating vineyard models opposite to each other at a realistic row width of 2.5 m. The ground was a flat surface with friction characteristics similar to soil. The simulation environment can be seen in Figure 1.

### 2.2. Simulated Robots

The simulated robots were the Robotnik Vogui robots [33]. These are the robots that are to be used in the eventual field experiments. The libraries provided by Robotnik include the built-in tools to simulate up to three robots in the Gazebo simulation environment. In this study, a two-robot simulation is presented, with an expert robot and a helper robot. The expert robot possesses a Universal Robots UR-5 arm equipped with an OnRobot RG2 gripper. On the other hand, a larger UR-10 arm of Universal Robots with a Schunk EGH gripper is mounted on the helper robot. The expert robot’s role is to perform the actual harvesting operations (detection and picking of grapes), while the helper robot assists the expert robot by carrying the grapes that the expert robot harvests. This is the reason why the helper robot is equipped with an arm with a larger payload compared to the smaller UR-5 arm mounted on the expert robot. In addition, the helper robot is equipped with a ZED mini stereo camera mounted on the end effector of its UR-10 arm, used for marker detection. Correspondingly, an 11 cm × 11 cm ArUco marker was placed at the back of the expert robot. ArUco markers, provided by the OpenCV library, are binary square fiducial markers consisting of a binary matrix code to allow identification and a black border which enables fast detection. They are extensively used for camera-based pose estimation because of fast and reliable detection.

Both robots are equipped with TiM511 2D LiDAR sensors located at the front of each robot and have a scanning range of 0.05 m to 10 m and a scanning frequency of 15 Hz. The horizontal aperture angle of these sensors is 180°, and scanning occurs at angle increments of approximately 0.22°.

Figure 2 illustrates the two simulated robots.

The robots run on the Robot Operating System (ROS) operating system on Ubuntu 18.04 machines. Apart from the preinstalled nodes provided by Robotnik for the operation of the Vogui robots, various ROS nodes have also been developed for the purposes of this work that implement the various functionalities. These additional nodes operate together with the preinstalled framework. Figure 3 shows the nodes developed that run on each robot and the connections between them. For clarity, the nodes provided by Robotnik for the Vogui robots are omitted. The operation of the nodes in Figure 3 is explained next.

The entire robot system is controlled by the Controller node, which is responsible for communications with the base station and with other robots for gathering the statuses of the various devices and for initiating and stopping the task itself by sending the relevant commands to the Task node. More specifically, the Task node is directly controlled by the controller node and performs the actual algorithm for the prescribed task. It contains the necessary functions that enable the node to send commands to the subsystems necessary for the completion of the task, such as navigation, arm movement, etc. These commands are issued in the order specified by the task. The Navigation node is responsible for moving the robot in various poses given appropriate commands from the Task node and by taking into account the relevant position sensors information. Similarly, the Arm node is responsible for performing the necessary arm and gripper movements according to Task node commands. The Vision node controls the camera and returns the appropriate data when requested by the Task node. In this paper, the Vision node is active only on the helper robot and performs ArUco marker pose estimation through the use of the aruco_detect ROS package. Finally, the MQTT/ROS bridge node translates ROS messages into MQTT format and vice versa.

### 2.3. Base Station

In addition, a base station was implemented, which serves two purposes: the first is to select and display the map of the field. The second is to continuously receive information regarding the status of the task and the robots. In the graphical user interface of the base station’s software, the user can initially select and load the desired vineyard map. The user can then design the robot’s desired path and the work areas on that path using a user interface, as seen in Figure 4. The path is designed by selecting waypoints on the map, at arbitrary distances between them. There are two types of available waypoints: navigation waypoints and work waypoints. The difference between the two is that the latter signifies the waypoints where the robots must stop and perform the desired task.

The selected waypoints on the map image are converted into coordinates in the world coordinate system. The user can then send the planned path to a robot as a list of coordinates. Each waypoint is also accompanied by a flag, say *f_work_*, indicating whether the waypoint is a work waypoint, i.e., a position where the expert robot must stop and perform its task, which in this case is harvesting.

When designing the path, the waypoints need to represent a feasible trajectory, facilitating navigation. For this reason, the waypoints need to be positioned such that they ensure smooth turning before entering a vineyard row. Also, the waypoints need to be spaced apart enough in order to minimize robot spatial interactions, unless they need to collaborate to carry out harvesting. Finally, waypoints within a vineyard row have to be positioned approximately in the center of the row.

The second purpose of the base station is to establish a communications server that handles bidirectional robot-to-base and robot-to-robot communications. The former is used so that the base station can send messages to the robots and also to receive messages from the robots when they periodically report their status. The robot status messages include their position and orientation, the status of the various subsystems, sensor measurements, and the progress of the task. The latter, i.e., robot-to-robot communications, is handled by assigning the base station to act as intermediary for robot-to-robot messages. This means that messages that are sent from one robot to another are first delivered to the base station, so that the base station logs the message, which is then forwarded to the receiver robot. Communication between the base station and the robots is achieved using the MQTT communications protocol.

### 2.4. Localization and Navigation

The methods employed to provide individual robot localization and motion planning, as well as the algorithms to ensure coarse robot positioning within a vineyard row are described in this section. These methods are proven and widely used in the literature. In addition, Adaptive Monte Carlo Localization (AMCL) and Timed Elastic Band (TEB) motion planning modules are provided and calibrated by the manufacturer specifically for the Vogui robots.

Localization for each robot is achieved using the AMCL system [34,35], which uses a particle filter to determine the pose of the robot given a known map. Usually, this implies an indoor environment, such as a static and structured environment that can be represented by a map, but this is also the case with a vineyard field with its well-defined rows. The map is supplied by the user based on a prior vineyard mapping processes using drones, during which aerial photos are obtained. The aerial photos are converted into maps which can then be imported into the robot’s localization module prior to the task, as well as loaded on the base station’s software as it was described in the previous section. The internal map allows for the computation of a global cost map, which determines the regions in which the robot is allowed to navigate. At the beginning of the task, the robots are positioned on the vineyard at the same position and orientation as the pose that is set in terms of the initial parameters on the AMCL module. During the execution of the task, the robots continuously estimate their location by comparing the LiDAR readings to the preloaded map. More specifically, the AMCL module utilizes the front LiDAR of the Vogui robots.

As mentioned earlier, the user plans the desired robot path on the map via the base station’s graphical user interface. The desired robot path is a collection of waypoints. Each waypoint consists of a coordinate pair on the 2D plane and an additional flag, *f_work_*, indicating whether the waypoint is inside a work area. A work area is defined as the area in which the expert robot must perform harvesting. Each robot moves from waypoint to waypoint by calculating a trajectory using the TEB method [36]. This method considers the local cost map as it is generated by the obstacle detection sensors (in this case the LiDAR sensors) and calculates optimal trajectories in real time. It supports differential drive as well as omnidirectional robots, as is the case with the Vogui robots used in this simulation study. The orientation of the robot when it arrives at the target waypoint is calculated to be equal to the slope of the line segment defined by the previous and the target waypoint.

Finally, a robot needs to correct its pose relative to the vineyard rows on its sides. For this operation, a method that utilizes the 2D LiDAR sensor was developed. The LiDAR continuously measures the distances to the objects each beam is reflected on and publishes this information to the system. When the navigation node is required to correct the robot’s pose, then the last LiDAR scan is stored and is processed as follows: the measured distances are partitioned into measurements on the left and on the right side of the robot. From these distances, and given the location of the LiDAR sensor (assumed to be at the origin of the local reference frame), the coordinates of the beam hits are calculated. Then, the RANSAC algorithm [37] is applied to these points in order to estimate the parameters of a linear model which describes each row. An example of this process is illustrated in Figure 5. With the LiDAR sensor located at (0, 0) and the robot facing to the right of the image, a map of the various detected locations can be constructed (red and blue crosses). The RANSAC algorithm produces the linear models of the vineyard rows on the left and right side of the robot (red and blue dashed lines, respectively). From the parameters of the linear model, the slope of the rows on the left and right sides of the robot can be determined and then compared to the robot’s orientation, which is at zero in this local coordinate system. The error angle (i.e., the angle that corrects the robot’s orientation to achieve a direction parallel to the row) is defined as the mean between the slopes of the linear models on the left and right sides of the robot.

It can be seen in Figure 5 that the detected locations are not producing an accurate line on either side because the vineyard does not have a smooth surface due to varying foliage density. This causes the two linear models to not be precisely parallel, and this is why the mean of the two slopes is considered. In addition, it can be seen that laser beams penetrate the crop rows and detect the vineyard rows in the adjacent corridors. However, because the density of beam hits is much higher in the vineyard rows that define the corridor the robot is moving in compared to the beam hits in rows of other corridors, the RANSAC algorithm treats the latter as outliers and calculates the linear model considering only the former. Also, the distance between the linear models on the left and right sides of the robot can be calculated, and the robot can move either to the center of the corridor or approach a vineyard row according to the task’s needs. The distance between the robot and the vineyard row on either side is defined as the shortest distance between the LiDAR and the line produced by the linear regression model.

### 2.5. Coordinated Robot Navigation

While the individual robots navigate by planning their motion from waypoint to waypoint, as described in the previous section, they also need to coordinate their motion so that they move in an orderly fashion, maintain control of the flow of the task, and ensure that the robots position themselves correctly in preparation for collaborative task execution. For this reason, a modular coordination strategy has been developed. The strategy employs the localization and navigation modules described in the previous section. The proposed strategy consists of two algorithms, one for the expert and one for the helper robot. It has been designed to ensure the correct operation in all phases of the task, including navigation from the initial position to entering a vineyard, navigation inside the vineyard rows, and the positioning of the robots relative to each other in preparation for the actual harvesting task. At the same time, in order to ensure safe operation, it was decided that the motion would be strictly regulated with the two robots moving alternately. To implement this, the two algorithms operating on each of the robots are triggered at various points using the communication module. The navigation algorithms for the expert and helper robots are illustrated in Figure 6.

Initially, the expert robot proceeds to the first waypoint it has received from the base station. When the expert robot arrives at the next waypoint, it checks the additional *f_work_* flag that accompanies the coordinates of the waypoint. If the *f_work_* flag is not set, then the expert robot proceeds to the next waypoint and sends its *previous* location to the helper robot. Upon receipt of this message, the helper robot moves to the transmitted location and notifies the expert robot of the completion of its move, which allows the expert robot to proceed to the next waypoint. On the other hand, if the *f_work_* flag is set, it means that the expert robot needs to stop, adjust its orientation so that its longitudinal axis is parallel to the row, and correct its lateral position so that its robotic arm can reach and harvest the grapes, on the robot’s side. When this positioning is completed, the expert robot then transmits its *current* pose together with a relevant command, which forces the helper to position itself behind the expert robot. The helper robot then approaches the expert robot from behind so that it stops at a predefined distance, *d_R_*, with the same orientation as the expert robot, as illustrated in Figure 7.

The calculated (*x_H_*, *y_H_*) position of the helper robot is such that it is a point on the axis of movement of the expert robot, which is defined by its position (*x_E_*, *y_E_*) and its orientation *h_E_*. The orientation, *h_H_*, of the helper robot must also be equal to that of the expert robot, as it has been previously adjusted to be parallel to the row.

### 2.6. Fine-Tuned Positioning

With the expert robot coarsely positioned and ready to harvest, it is necessary that the helper robot is correctly positioned behind it. The helper robot initially approaches the expert robot as described previously based on the location and heading measurements transmitted by the expert robot. However, the accuracy of these measurements is not adequate for the precise manipulation actions between the two robots for the following reason: location and heading estimates are produced by the comparison between the internal map and the LiDAR measurements. Although LiDAR sensors are adequately accurate, the localization accuracy is not because of foliage variations. Due to this uncertainty, it is likely that the helper robot may be initially positioned too far from the expert robot for its UR-10 arm to be able to reach and manipulate the expert robot’s basket effectively. Because the ability to manipulate the basket must be guaranteed, the fine-tuning of the helper robot’s positioning is required. This is an additional module of the proposed coordination strategy, and it is a part of the helper robot’s algorithm, as seen in the flowchart of Figure 6b. This is achieved through the ArUco marker located at the back of the expert robot and its detection by the Zed mini camera on the helper robot. After the initial approach, the marker is clearly visible and detectable. Its pose is calculated by the aruco_detection ROS package and made available to the system. This information is utilized by the navigation node, which moves the robot in a way that the desired positioning is achieved and, therefore, makes the basket reachable by the UR-10 arm of the helper robot. When this corrective motion is finished, and basket manipulation is required, the marker’s pose can again be sampled in order to calculate the position of the basket relative to that of the marker. 

Figure 8 shows the view from the helper robot’s camera during the detection of the expert robot’s ArUco marker.

The superimposed axes shown on the marker in Figure 8 represent the detected pose of the marker in 3D space. The *z*-axis (the blue axis at the center of the marker in Figure 8, with direction out of the page) of the detected pose is used to correct the helper robot’s positioning. This is the distance between the robot’s camera and the marker. A suitable approach distance was determined to be 40 cm. At this distance, the marker pose detection is reliable and the UR-10 arm is able to reach the basket.

## 3. Results

Simulation experiments were conducted to test the effectiveness of the various algorithms described above and their behavior when applied in the complete collaborative navigation strategy. The experimental design was focused on evaluating the following aspects of the strategy:The robustness of the strategy;The evaluation of the localization system;The effectiveness of vision-based fine-tuned positioning;The effect of localization accuracy in coordinated robot navigation;The effect of waypoint selection on localization accuracy.

The experiments involve the two robots navigating a row within the vineyard based on a specific path that was previously generated by the base station software. The path is shown in Figure 9. The starting and the end positions of the robot are also indicated.

As can be seen in Figure 9, the path includes both a trajectory outside a vineyard row (red dots) as well as work waypoints where harvesting must occur (green dots), i.e., where the robots must position themselves properly in preparation for the harvesting task. The work waypoints are positioned to be equidistant from the plant rows so that the maximum spacing of the robots from the plants is achieved. The waypoints which are outside the vineyard rows are selected such that the correct turns towards the vineyard row are ensured. Finally, where possible, the waypoints are spaced far apart in order to prevent the robots from stopping frequently or approaching each other unless it is necessary for the harvesting task. Therefore, this path is suitable for testing all aspects of the navigation algorithm presented in the previous section. It must be noted that the selected path pertains primarily to the expert robot since the expert robot directly follows the path, whereas the helper robot visits locations that the expert robot transmits. This is why the initial position shown in Figure 9 is actually the expert robot’s initial position.

Since the main objective of the study was the collaborative navigation aspect and not the actual harvesting task, the latter was replaced by a simplified arm movement sequence as follows: when the robots are correctly positioned for harvesting, the expert robot performs an arm movement. When complete, the helper robot detects the marker and approaches the basket with its UR-10 arm.

Ten trials of the experiment using the path of Figure 9 were carried out. In all trials, the aforementioned task was successfully completed. During the trials, the actual robot positions and orientations were recorded at each waypoint. The distance between the actual position and the desired position represents the navigational error at each waypoint. In order to determine the navigational error for the entire trajectory, the mean navigational error of all waypoints for each robot was calculated. Similarly, the difference between the actual and desired orientation for each waypoint was calculated, and the mean orientation error for the entire trajectory was determined. These calculations were performed for all ten trials. The results are summarized in Table 1.

As seen in Table 1, the mean errors for both the expert and the helper robots are comparable. Note that, even though the observed position errors are not particularly small, they are acceptable given that the task was completed successfully, without collisions. The orientation errors, on the other hand, are very small. The lack of accuracy can be explained by discrepancies between the map loaded into the robot and the environment that was actually perceived by the LiDAR sensors. In particular, while the vines are represented as solid walls with smooth surfaces on the map (see, for example, Figure 9), the virtual environment consists of complex three-dimensional vineyard models with irregular surfaces and empty regions (Figure 8).

Furthermore, positioning accuracy when using the camera and the ArUco marker was investigated (fine-tuned positioning). In the experimental scenario, out of all waypoints that constitute the robot’s trajectory, there are three work waypoints (denoted W1, W2, and W3) where the robots needed to be accurately positioned relative to each other in preparation for the harvesting operation (i.e., waypoints where the *f_work_* flag is set). In these instances, the target distance between the camera of the helper robot and the marker attached to the expert robot is set to 40 cm. For each trial, the actual distance achieved was recorded after every fine-tuned positioning, and the mean error of all three waypoints was calculated. The results are shown in Table 2.

The results presented in Table 2 show that the fine-tuned positioning ensures that the helper robot is accurately positioned behind the expert robot, regardless of the inaccurate initial approach. This means that the accurate and reliable positioning of the UR-10 arm is feasible with respect to the expert robot’s basket, and this allows the collaborative aspect of the task to be carried out, i.e., the transfer of the basket from one robot to the other using the manipulator of the helper robot. More specifically, the accuracy achieved in positioning the robot is approximately 1 cm. 

It should be noted that, during the aforementioned experiments, when the expert robot transmitted the location of a waypoint to the helper robot, the coordinates used were the ones received from the base station, i.e., the ideal location of the transmitted waypoint as selected by the user. However, as seen in Table 1, there is an offset between the desired and the actual expert robot position at each waypoint. The second set of experiments was carried out in order to investigate whether any differences in performance can be observed when the expert robot communicates its actual coordinates to the helper robot at each waypoint, i.e., to determine the effect of localization accuracy in the coordinated robot navigation. The results are shown in Table 3 and Table 4 below.

A comparison between the results presented in Table 1 and Table 3 reveals that the expert robot follows the waypoints as accurately as before. This is expected since the expert robot follows the same waypoints as before. However, it is observed that the helper robot’s movement is slightly less accurate when it follows the actual position of the expert robot than when it follows the desired position of the expert robot. There is no significant difference in terms of orientation. Accurate fine-tuned positioning is also achieved in this case.

The final aspect of the proposed strategy’s evaluation was aimed at investigating the effect of waypoint selection on the accuracy of the navigation of the two robots. In order to test this aspect, a second path was designed, and in this case, the path consisted of denser waypoints as the robots approached the vineyard corridor. This second path is illustrated in Figure 10 below.

Ten trials were executed using this new test path and the average position and orientation errors were calculated as before. The results are summarized in Table 5. 

From Table 5, it can be deduced that the use of more waypoints in the design of the robot’s path results in a slightly more accurate waypoint following for the expert robot. This can be attributed to reduced error accumulation between the waypoints. However, the orientation errors slightly increased. Overall, the different approaches in the communication of the coordinates and waypoint selection yielded significant differences, both in terms of robustness as well as in terms of accuracy.

## 4. Discussion

In this paper, a modular collaborative navigation strategy for an agricultural task is proposed. The strategy involves two robots within a vineyard: a leader and a follower, communicating their position in order to coordinate their movement on a predefined path designed on separate software installed on a base station. The strategy employs well-established localization and planning modules that are executed at the individual robot level. Localization is achieved using both a preloaded map and LiDAR sensors. Additional task-specific algorithms were developed as part of the overall strategy, including vineyard row parallelization and the precise positioning of the two robots relative to each other in order to achieve a collaborative task. The methods described in this paper were developed in a modular fashion, which means that they can be selected and combined to be used in both single-robot tasks as well as in multirobot scenarios, according to the task specifications. These methods form part of the coordination algorithms executed by the robots toward the successful completion of the prescribed task.

The proposed modular collaborative navigation strategy has been tested in a harvesting scenario, where two robots start from a predetermined known position in the vineyard and, following a predefined path, they traverse a vineyard row, stopping at predetermined positions to perform a collaborative harvesting task, in which one robot performs the harvesting and the other collects the harvested grapes. Different variations of the strategy have been investigated, and the performance quantified. The successful completion of all experimental trials suggests that the proposed strategy is robust and effective in guiding the robots in completing the collaborative task. Even though the localization is not optimal, the existing coarse navigation is adequate for traversing the narrow vineyard rows. Localization errors occur during navigation due to wheel slippage (causing odometry errors) or the inherent inaccuracies in the localization method. However, where localization precision is required, in particular for the positioning of the two robots relative to each other during harvesting, a vision-based method is employed in order to reduce any errors after the initial approach. This would also apply in the case of the real harvesting task on the part of the expert robot, where the robot would coarsely position itself parallel to the row, and then vision would accurately guide the robotic arm toward the grapes to be harvested. 

The modular nature of the proposed strategy means that any of the individual methods for localization, navigation, positioning, etc., that are involved in the workflow can be replaced by others depending on the available sensors. This implies that the strategy is not robot-specific and can be applied to robots with different hardware specifications. Another advantage of the strategy is its scalability since it allows for the use of more than two robots to be integrated into a task using the same communications protocol and work sequence.

## 5. Conclusions

In this paper, a strategy for two cooperative and mobile robots for harvesting a vineyard is described. The two robots, the expert and the helper, were tasked with accurately navigating the corridors of a vineyard, while the helper robot needed to position itself accurately behind the expert robot at a predefined distance with the help of a vision system in order to complete a specific task. The simulation results presented indicate that the proposed coordination strategy is sound, and therefore, future work will involve implementation and experimentation with real robots in order to validate these results in the field. Because of the nature of the simulation studies, which involved the use of simulated ROS-based systems, the developed software is expected to be highly portable to the actual systems. In addition, further work is needed to improve the accuracy of the localization, which will include the use of other sensors, such as GPS modules, and more accurate map representations. Additionally, in the final system, the intent is that the waypoints are generated automatically when given a vineyard map. More work is also needed to evaluate the performance of the proposed strategy with respect to the positioning of the waypoints. Finally, future work will report on the implementation of novel intelligence models [32] on agrobots regarding skillful agricultural tasks.

## Figures and Tables

**Figure 1 sensors-22-09095-f001:**
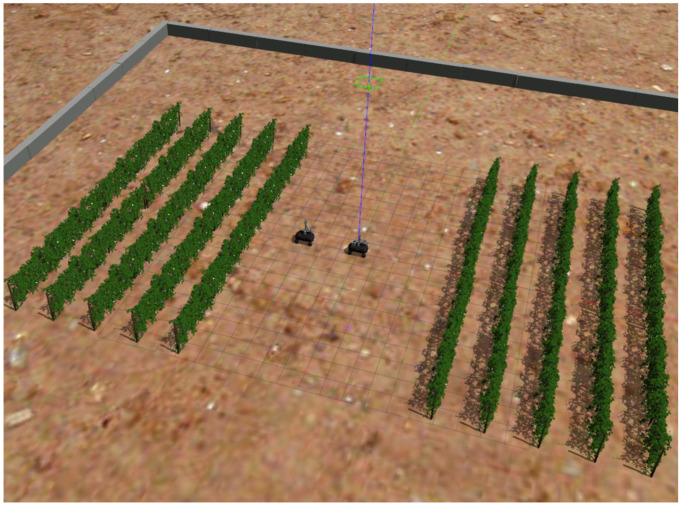
The simulated vineyard.

**Figure 2 sensors-22-09095-f002:**
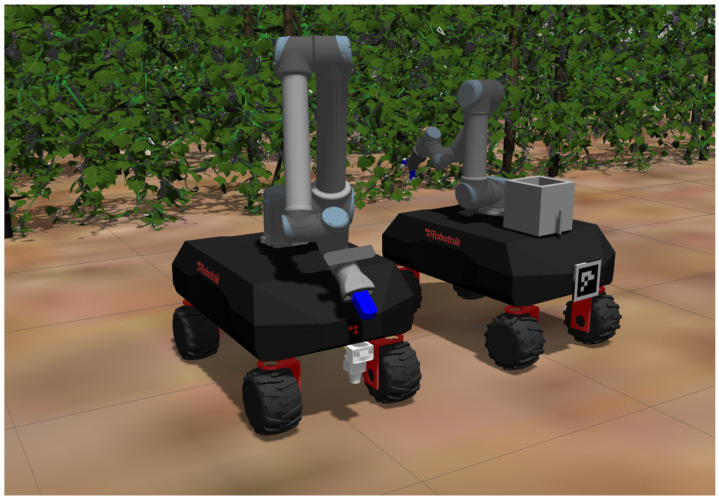
The simulated expert (**right**) and helper robots (**left**).

**Figure 3 sensors-22-09095-f003:**
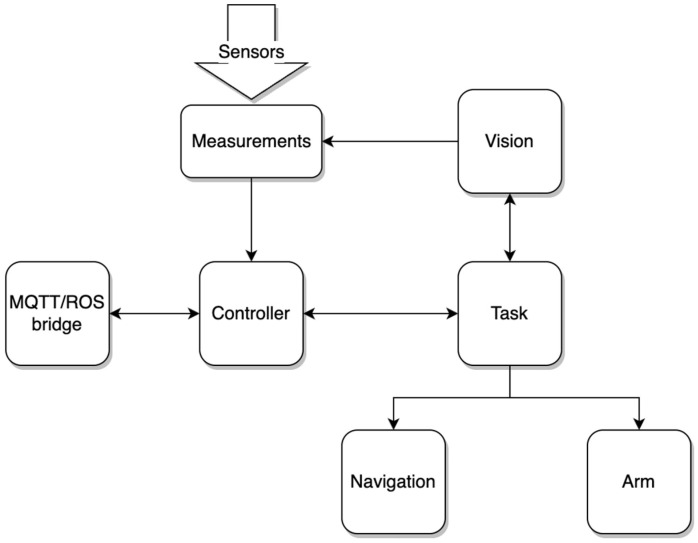
Additional nodes running on each robot. The arrows indicate how the nodes communicate with each other.

**Figure 4 sensors-22-09095-f004:**
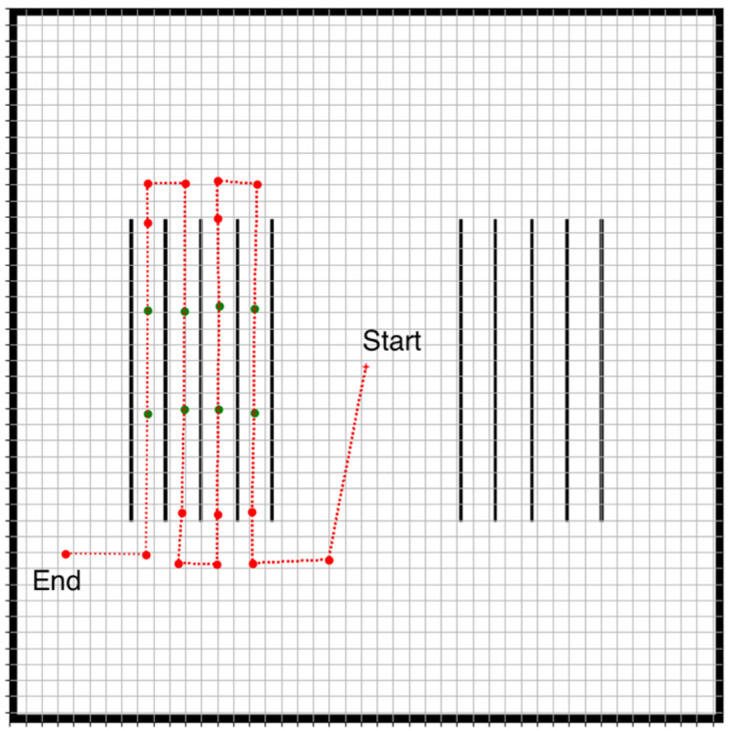
Path selection using the base station’s user interface. The red dots are the selected waypoints, and the red line indicates the desired robot path. The green dots are work waypoints where the robots must stop and harvest. The red cross indicates the initial position of a robot. The black lines are the vineyard rows.

**Figure 5 sensors-22-09095-f005:**
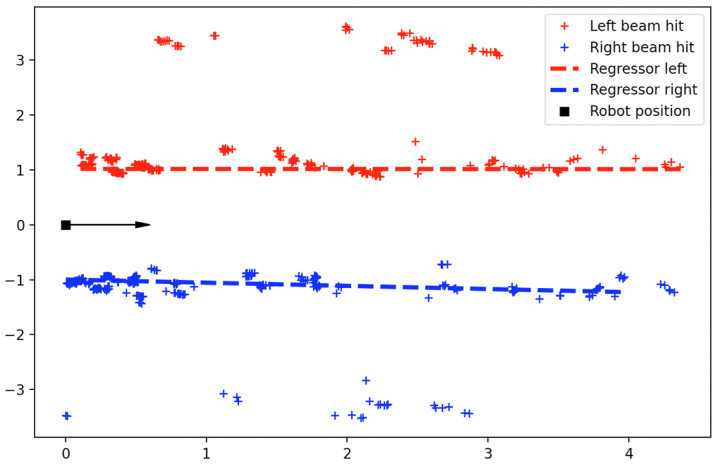
Vineyard row detection using a LiDAR scan.

**Figure 6 sensors-22-09095-f006:**
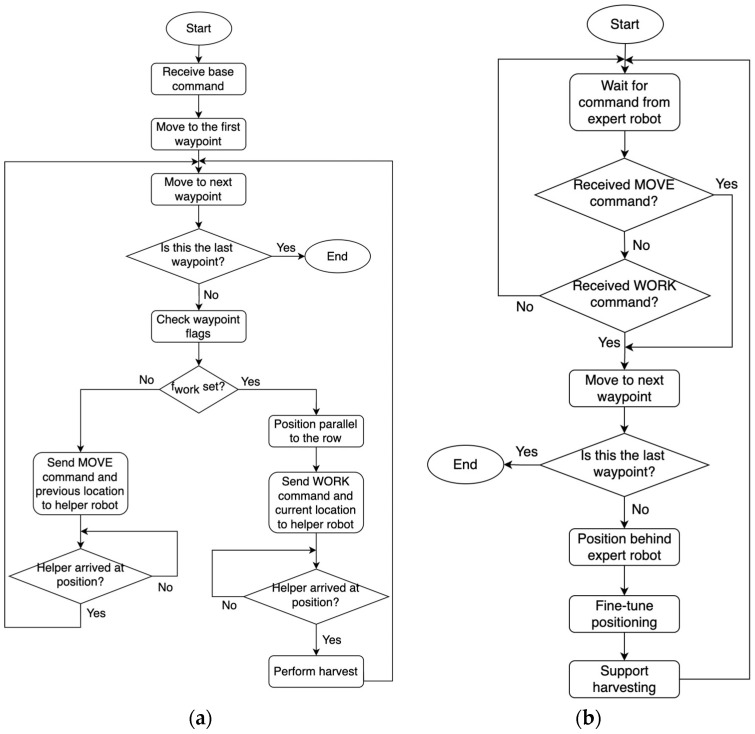
Coordinated navigation algorithms for (**a**) the expert robot and (**b**) the helper robot.

**Figure 7 sensors-22-09095-f007:**
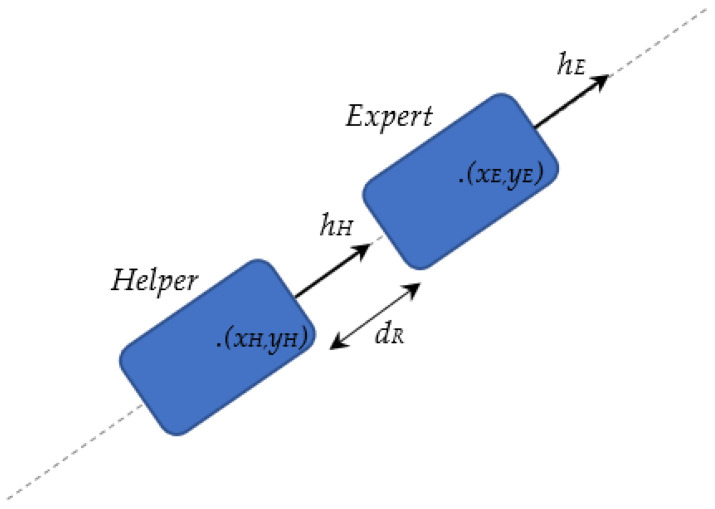
Initial positioning of the helper robot behind the expert robot.

**Figure 8 sensors-22-09095-f008:**
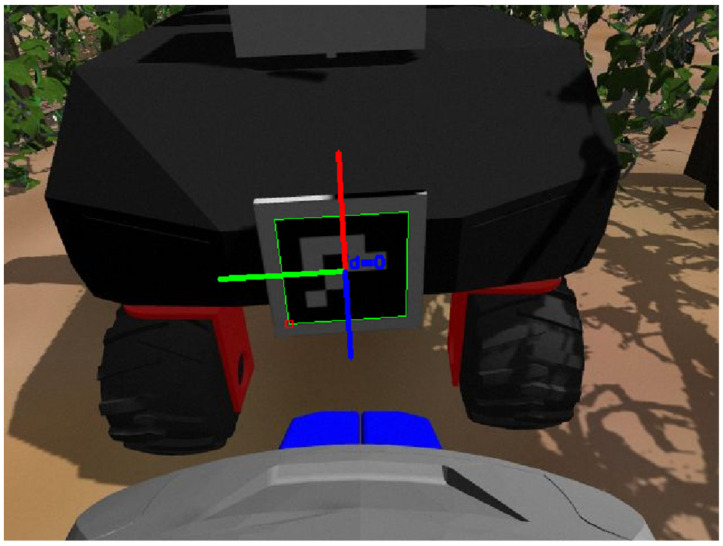
Fine-tuning the positioning of the helper robot behind the expert robot for task initiation using the ArUco marker. View from the helper robot’s camera.

**Figure 9 sensors-22-09095-f009:**
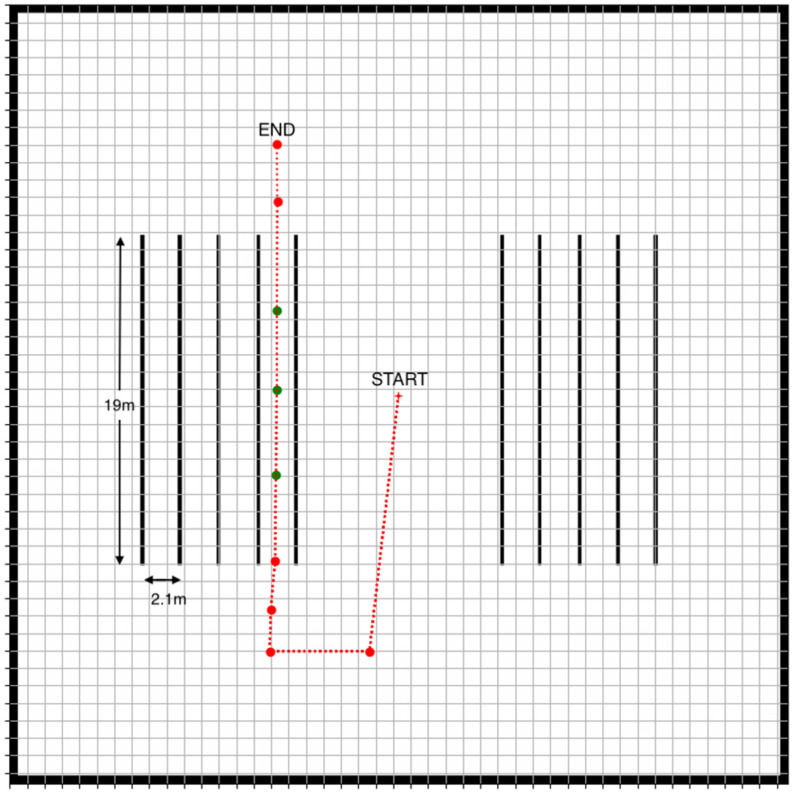
The test robot path used in the experiments. The red dots are the selected waypoints, and the red line indicates the desired robot path. The green dots are work waypoints where the robots must stop and harvest. The red cross indicates the initial position of a robot.

**Figure 10 sensors-22-09095-f010:**
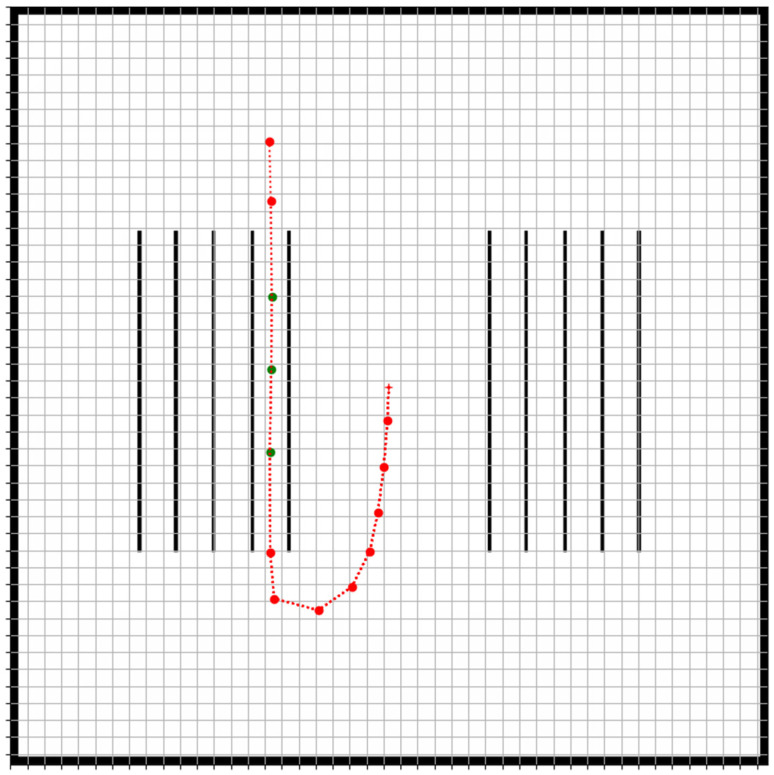
The second test robot path with denser waypoints. The red dots are the selected waypoints, and the red line indicates the desired robot path. The green dots are work waypoints where the robots must stop and harvest. The red cross indicates the initial position of a robot.

**Table 1 sensors-22-09095-t001:** Position and orientation errors for the expert and helper robots for the first test path.

Trial	Expert Robot Mean Trajectory Error (m)	Expert Robot Mean Orientation Error (°)	Helper Robot Mean Trajectory Error (m)	Helper Robot Mean Orientation Error (°)
1	0.259	2.356	0.238	1.645
2	0.266	2.489	0.256	2.619
3	0.253	2.316	0.219	4.453
4	0.259	2.169	0.213	1.994
5	0.265	2.150	0.208	2.840
6	0.251	2.351	0.278	2.965
7	0.271	2.590	0.262	2.967
8	0.233	2.029	0.242	2.607
9	0.230	2.492	0.229	3.305
10	0.259	1.844	0.250	5.162
**Mean error**	**0.255**	**2.279**	**0.239**	**3.056**

**Table 2 sensors-22-09095-t002:** Position errors measured during fine-tuned positioning for the first test path.

Trial	Waypoint	Distance at First Approach (m)	Distance after Fine-Tuning (m)	Error (m)
1	W1 W2 W3	0.751 0.628 0.693	0.427 0.423 0.424	0.024
2	W1 W2 W3	0.498 0.500 0.618	0.420 0.422 0.435	0.026
3	W1 W2 W3	0.908 0.701 0.657	0.407 0.413 0.423	0.014
4	W1 W2 W3	0.769 0.785 0.589	0.409 0.413 0.403	0.008
5	W1 W2 W3	0.787 0.851 0.638	0.416 0.420 0.418	0.018
6	W1 W2 W3	0.590 0.710 0.734	0.416 0.424 0.423	0.021
7	W1 W2 W3	0.718 0.671 0.697	0.413 0.400 0.424	0.012
8	W1 W2 W3	0.781 0.710 0.526	0.424 0.429 0.414	0.023
9	W1 W2 W3	0.527 0.508 0.444	0.409 0.418 0.412	0.013
10	W1 W2 W3	0.896 0.957 1.001	0.423 0.410 0.422	0.018
**Mean error**				**0.018**

**Table 3 sensors-22-09095-t003:** Position and orientation errors for the expert and helper robots for the second test path.

Trial	Expert Robot Mean Trajectory Error (m)	Expert Robot Mean Orientation Error (°)	Helper Robot Mean Trajectory Error (m)	Helper Robot Mean Orientation Error (°)
1	0.262	2.005	0.258	2.313
2	0.231	2.007	0.249	2.204
3	0.234	1.641	0.253	2.726
4	0.261	2.164	0.246	2.807
5	0.263	2.039	0.261	2.966
6	0.247	2.649	0.270	2.632
7	0.264	2.188	0.252	2.911
8	0.254	2.192	0.217	3.505
9	0.280	2.184	0.242	1.904
10	0.251	1.959	0.265	2.695
**Mean error**	**0.255**	**2.103**	**0.251**	**2.666**

**Table 4 sensors-22-09095-t004:** Position errors measured during fine-tuned positioning.

Trial	Waypoint	Distance at First Approach (m)	Distance after Fine-Tuning (m)	Error (m)
1	W1 W2 W3	1.107 1.042 1.141	0.399 0.500 0.470	0.056
2	W1 W2 W3	0.858 0.838 0.760	0.421 0.400 0.404	0.008
3	W1 W2 W3	0.923 0.971 0.827	0.410 0.411 0.419	0.013
4	W1 W2 W3	0.988 0.986 1.034	0.421 0.419 0.526	0.055
5	W1 W2 W3	0.898 0.832 0.681	0.404 0.405 0.404	0.004
6	W1 W2 W3	1.101 0.986 1.195	0.408 0.406 0.450	0.021
7	W1 W2 W3	1.082 1.141 1.057	0.449 0.509 0.449	0.069
8	W1 W2 W3	0.673 0.666 0.531	0.455 0.465 0.423	0.048
9	W1 W2 W3	1.010 1.119 1.040	0.447 0.443 0.453	0.047
10	W1 W2 W3	0.961 0.855 0.920	0.441 0.448 0.457	0.049
**Mean error**				**0.037**

**Table 5 sensors-22-09095-t005:** Average position and orientation errors for the expert and helper robots for the second test path.

Trial	Expert Robot Mean Trajectory Error (m)	Expert Robot Mean Orientation Error (°)	Helper Robot Mean Trajectory Error (m)	Helper Robot Mean Orientation Error (°)
1	0.221	3.137	0.230	3.522
2	0.220	2.051	0.271	3.145
3	0.231	3.330	0.251	3.333
4	0.243	5.851	0.265	3.364
5	0.236	4.583	0.248	3.164
6	0.235	6.101	0.222	2.550
7	0.206	4.785	0.266	3.854
8	0.217	4.603	0.264	4.320
9	0.235	5.816	0.231	1.548
10	0.246	3.850	0.280	4.281
**Mean error**	**0.231**	**4.714**	**0.251**	**3.416**

## Data Availability

Not applicable.

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
