# Peer review of "Coordinated Navigation of Two Agricultural Robots in a Vineyard: A Simulation Study"

_sensors, 2022, doi:10.3390/s22239095_

Round 1
Reviewer 1 Report
The paper presents a collaborative navigation strategy for an agricultural task. The strategy involves two robots, a leader and a follower, communicating their position in order to coordinate their movement on a predefined path installed on a base station within a vineyard. Localization is achieved using both a pre-loaded map and LiDAR sensors. Task-specific algorithms have been developed, including vineyard row parallelisation and precise positioning of the two robots relative to each other in order to achieve a collaborative task.
The proposed system contemplates two robots, an expert robot, responsible for performing harvesting, and a helper robot, tasked with supporting the expert robot by carrying and transporting the harvested grapes, are navigating collaboratively in a vineyard. The robots must be able to position themselves in such a way so that precise manipulation is possible.
The expert robot (with a UR5 arm, a OnRobot RG2 gripper and a LIDAR) starts from a predetermined known position in the vineyard and, following a predefined path, they traverse a vineyard row, stopping at pre-determined positions to perform a collaborative harvesting task. The expert robot begins harvesting and deposits the harvested grapes to a basket located at the back of its chassis. When the basket is full, the helper robot (With a UR10 arm, a Schunk EGH gripper, a LIDAR and a ZED mini stereo camera) picks up the basket and deposits its contents to a larger basket which, if full, needs to be emptied at another location, while the expert robot remains at its current position. The Vision node is active only on the helper robot and it is performing ArUco marker pose estimation.
Experiments shown he successful completion of the developed system and suggest that the proposed strategy is robust and effective in guiding the robots in completing the collaborative task.
General comments:
- The system contemplates two robots that need to self-position. In the description of the algorithms, there is no novelty regarding the self-positioning systems.
- Do the UR5 and UR10 arms have cameras? With regard to precision, a good sensorization of the arms is more useful than a fine adjustment of the position of the bases.
- The main contribution is the collaborative navigation strategy algorithm. It seems to have no more complexity than timing. The difficulties regarding the problem in the agricultural task and how it seeks to solve them should be better explained.
- You said it is necessary that the helper robot is correctly positioned behind it. ¿Is it really necessary for the task?
Other comments:
- Explain better your algorithms and contributions.
- Figure 3 it is too much general, does not show any novelty unless the particularities of each node to the proposed system are described.
- Improve figure 4 an its explanation. How do waypoints and the harvesting position affect the multi robot collaboration system?
- Figure 5, as figure 3, it is too much general, does not show any novelty unless the particularities of each node to the proposed system are described.
- Improve figure 10 and its explanation about how they affect the multirobot collaboration system.
- - More experimental results and comparatives must be included.
Reviewer 2 Report
1. Literature review needs to include several recent, relevant publications (high impact) highlighting their key findings. The current version only discussed general aspects while reviewing each from several papers is necessary. You may provide a review summary table consisting of a column for the comments or key conclusions.
2. Enhance the objective and novelty of the work in the introduction section.
3. Separate the Conclusion from the Discussion section.
4. Improve language throughout the manuscript.
Reviewer 3 Report
This paper presents a simulation study that demonstrates a robust coordination strategy for the navigation of two heterogeneous robots, where one robot is the expert and the second robot is the helper, in a vineyard. The robots are equipped with localization and navigation capabilities so that they can navigate the environment independently and appropriately position themselves in the work area. A modular collaborative algorithm is proposed for the coordinated navigation of the two robots in the field, via a communications module. Furthermore, the robots are also able to accurately position themselves relative to each other, using a vision module, in order to effectively perform their cooperative task. For the experiments, a realistic simulation environment is considered and the various control mechanisms are described. Experiments are carried out to investigate the robustness of the various algorithms and provide preliminary results before real life implementation.
However, there are a few Major observations based on which this paper cannot be accepted in its current form until it is improved. The comments are as follows.
1.1.The work presented by authors seems like a summary of how different open source applications/simulations were integrated and executed.
2.2. There is no novelty in the presented work.
33. Authors have used different open source applications e.g. Gazebo simulation , Robotnik libraries ,and AMCL system etc. The author should have explained that why they selected these applications and what these have to offer more than other similar applications.
44. There should be a comparison of features of existing open source applications.
5. The survey lacks an evaluation of prominent state-of-the-art techniques. An experimental setup should be included to evaluate different techniques based on performance metrics.
6 6. The author may also categorize different techniques based on performance results and also identify the scenarios where these techniques can be applied.
7 7. An experimental setup diagram may also be included which can explain the evaluation model.
88. Results section can be improved by comparing the results with other existing methods.
Round 2
Reviewer 1 Report
The paper presents a collaborative navigation strategy for an agricultural task. The strategy involves two robots, a leader and a follower, communicating their position in order to coordinate their movement on a predefined path installed on a base station within a vineyard. Localization is achieved using both a pre-loaded map and LiDAR sensors. Task-specific algorithms have been developed, including vineyard row parallelisation and precise positioning of the two robots relative to each other in order to achieve a collaborative task.
The proposed system contemplates two robots, an expert robot, responsible for performing harvesting, and a helper robot, tasked with supporting the expert robot by carrying and transporting the harvested grapes, are navigating collaboratively in a vineyard. The robots must be able to position themselves in such a way so that precise manipulation is possible.
General Comments:
- The revision has been carried out exhaustively, considerably improving the publication.
- The contributions have been reinforced and highlighted and the explanations of the state of the art have been improved to highlight the contributions of the work, which, although fair, are sufficient to endorse the publication.
- Regarding technical contributions, the need for the two robots and their integration and the need for the positioning of the two robots have been highlighted. The description of the experiments and the added results support the proposal much better, as well as the analysis of the results.
- The discussion and the conclusion also better collect the contributions of the work.
Details:
- Figures and explanations have been improved enough.
Author Response
Thank you for your constructive review
Reviewer 2 Report
Accept
Author Response
Thank you for your constructive review
Reviewer 3 Report
The response file of authors is very frustrating and confusing, they dint bothered to mention the page nos and line nos where they did changes and left a puzzle for reviewer to do it himself. As a reviewer this thing is highly frustrating for me and i am returning this with a revision needed to properly mark where (pages. lines) and what changes they made with a detailed reasoning in response and not just the general shallow reasoning.
Author Response
Apologies for the previous incomplete response. The attached file now includes more detailed reasonings and page numbers and line numbers for the corresponding changes

Round 3
Reviewer 3 Report
Its accepted from my side.